# Protective Role of Natural Compounds under Radiation-Induced Injury

**DOI:** 10.3390/nu14245374

**Published:** 2022-12-17

**Authors:** Annamaria Altomare, Michele Fiore, Gabriele D’Ercole, Elena Imperia, Roberta Maria Nicolosi, Susanna Della Posta, Gabriella Pasqua, Michele Cicala, Laura De Gara, Sara Ramella, Michele Pier Luca Guarino

**Affiliations:** 1Unit of Food Science and Nutrition, Università Campus Bio-Medico di Roma, Via Alvaro del Portillo 21, 00128 Rome, Italy; 2Research Unit of Gastroenterology, Università Campus Biomedico di Roma, Via Alvaro del Portillo 21, 00128 Rome, Italy; 3NBFC, National Biodiversity Future Center, 90133 Palermo, Italy; 4Operative Research Unit of Radiation Oncology, Fondazione Policlinico Universitario Campus Biomedico di Roma, Via Alvaro del Portillo 21, 00128 Rome, Italy; 5Research Unit of Radiation Oncology, Department of Medicine and Surgery, Università Campus Bio-Medico di Roma, Via Alvaro del Portillo 200, 00128 Rome, Italy; 6Department of Environmental Biology, Sapienza Università di Roma, P.le Aldo Moro 5, 00185 Rome, Italy; 7Operative Research Unit of Gastroenterology, Fondazione Policlinico Campus Bio-Medico di Roma, Via Alvaro del Portillo 200, 00128 Rome, Italy

**Keywords:** radiotherapy, radioprotective, mucositis, radiation-induced mucositis, natural compounds

## Abstract

In recent years, evidence has shown the potential therapeutic effects of different natural compounds for the prevention and treatment of radiotherapy-induced mucositis (RIOM). RIOM represents one of the most frequent side effects associated with anti-neoplastic treatments affecting patients’ quality of life and treatment response due to radiation therapy discontinuation. The innate radio-protective ability of natural products obtained from plants is in part due to the numerous antioxidants possessed as a part of their normal secondary metabolic processes. However, oxygen presence is a key point for radiation efficacy on cancer cells. The aim of this review is to describe the most recent evidence on radiation-induced injury and the emerging protective role of natural compounds in preventing and treating this specific damage without compromising treatment efficacy.

## 1. Introduction

Even though anti-neoplastic treatment has become more effective, it continues to be associated with several short and long-term side effects [1]. Radiotherapy has variable success depending on the primary tumor [2], but the toxicity or side effects associated with its use can determine several disorders, such as up- and down-gastrointestinal (GI) mucositis, which affect the quality of life of patients and treatment response possibly leading therapy discontinuation [1]. In the GI tract, the epithelial cells have the highest rate of cell proliferation and turnover in the human body [3]; the direct and indirect biological effects of radiation therapy determine the production of reactive species and proinflammatory cytokines [4] responsible for the initiation of inflammatory processes that can affect cell proliferation and surveillance. In recent years, evidence has shown the potential therapeutic effects of several natural compounds for the prevention and treatment of radiotherapy-induced mucositis (RIOM) [5]. This review proposes an overview of the most recent evidence on radiation-induced injury and the emerging protective role of natural compounds in preventing and treating this specific damage, without compromising treatment efficacy.

The present review was based on a literature search in the databases PubMed, Scopus, and Google Scholar. The findings of research studies chosen from more than 900 viewed scientific publications were included/compared (about 80% of the articles were removed due to inconsistency regarding natural compounds’ role under radiotherapy). The search was based on keywords and phrases containing combinations of words such as radiotherapy, mucositis and natural compounds. In this article, we primarily include research findings from the years 2005 to 2022. 

## 2. Radiation-Induced Injury

Radiation therapy is one of the most widely used treatments for cancer worldwide. Approximately 50% of cancer patients receive radiation treatment [6], with about 3 million patients in Europe each year [7]. It can be used for different purposes: for curative intent with the aim to eradicate the disease, for neo-adjuvant intent (i.e., before surgery, to reduce the size of the disease and facilitate radical resections), for adjuvant intent (after surgery to eliminate any residual cells and reduce the risk of loco-regional recurrence) and finally for palliative purposes (such as reduction of pain or hemostatic effect). This treatment, even more when combined with systemic therapy (chemotherapy, target therapies, immunotherapy), can cause side effects that vary depending on the irradiated site. Healthy organs close to the radiation target are defined as organs at risk (OARs). Efforts are made to spare OARs during radiation treatment. A continuous update of the radiation techniques, dose prescription and fractionations changes in the treatment volume, and a sudden technological evolution allows for performing very precise radiation treatments with minimum dose dispersion to OARs. Despite these advances, when OARs were very close to the target, some radiation doses affected normal structures. Depending on the radiosensitivity of normal tissues and the dose delivered, several side effects can occur related to the irradiated organ. In radiation oncology, radiosensitivity is defined as the susceptibility to develop radiation-induced side effects [8]. Side effects also depend on the volume of healthy tissue exposed to radiation. Tissue damage caused by radiotherapy is generally locoregional and can be clinically divided into early and late side effects. Symptoms that occur in the acute phase, within a few days of starting radiotherapy treatment, are due to cellular damage. In tissues with high replicative turnover, such as skin, bone marrow, oral mucosa or intestinal mucosa, the tissue does not regenerate fast enough relative to the damage and may experience toxicity. These effects may include skin erythema, dry or moist desquamation of the skin, mucositis, nausea, diarrhea, edema or headaches. In addition, cellular injury causes a locoregional inflammatory reaction that contributes to the pathogenesis of the damage. The pathogenesis of late side effects is more complex. These effects, unlike acute ones, are generally progressive and irreversible. Tissues with a low replicative index instead, such as salivary glands, suffer more from side effects even months after treatment [9]. Late side effects include radiation-induced fibrosis, atrophy and vascular damage. Thus, acute and chronic side effects caused by radiation therapy follow different pathogenic pathways. Nevertheless, the severity of acute effects may have an influence on the eventual intensity of chronic effects (so-called consequential late effect). In order to reduce toxicities, it is important to understand the mechanisms underlying the radiation injury of normal tissues. In the tissue damage caused by radiotherapy treatments, regeneration processes, which are usually finely modulated, are altered. Often, supportive therapies are needed to help patients to treat symptoms.

## 3. Radiation-Induced Mucositis

The pathobiology of radio-induced mucositis was described by Sonis [10]: the damage process begins at the start of radiotherapy treatment. The first phase corresponds with the onset of mucositis, caused by direct and indirect damage. Damage at the level of the DNA double helix can result in cell damage or death. DNA helix breaks can, in fact, result in cellular damage or death. On the other hand, the effect of radiation on water molecules contained inside the cells can cause the formation of oxygen radicals: these cause significant damage to the molecules that are part of the cellular structures. Moreover, radiation can activate the transcription of factors (such as NF-κB and NRF2) that can stimulate the inflammatory response. In addition, there is a massive production of pro-inflammatory cytokines that increase tissue damage. Toxicities can seriously affect the course of cancer treatment and therefore the effectiveness of therapies. Mucositis is very frequent, enough to consider that radiation-induced oral mucositis occurs in about 80% of patients with head and neck neoplasia and in 100% of patients who undergo radiotherapy treatment with multiple daily sessions (hyper-fractionated) [11]. The way tissues respond to radiation is closely related to the intrinsic sensitivity of the cellular compartments that make up tissue (direct cell death) and the complex mechanisms of cross-talks that exist within these structures (indirect functional effects). Tissues with high cell turnover (such as epithelium, mucous membranes, bone marrow) suffer acute effects in response to radiation. This happens because cancer treatments cause severe depletion of stem cell compartments with high replicative turnover, which ultimately results in a loss of the supply of mature cells downstream. This process is accelerated in organs that undergo cell turnover faster than others [12].

## 4. From Pathogenesis to the Clinic of Mucositis

Radiation-induced damage to organs in the upper gastrointestinal (GI) tract results from the irradiation of GI neoplasms or tumors adjacent to it (e.g., head and neck, trachea, bronchi, mediastinal lymph nodes). This anatomical district, given the high cellular turnover, mostly suffers acute damage. The most common side effects range from loss of appetite to inflammation of the mucosa (e.g., esophagitis, duodenitis) and even ulcer and stenosis. Mucosal inflammation of the upper GI tract is often associated with an increased risk of infection by fungal agents, such as *Candida albicans*. From a radiobiological point of view, the type of damage that occurs at the mucosal level is deterministic: this means that as long as some “threshold” doses are maintained, no damage occurs. For reasons related to radiotherapy treatment planning, and due to the proximity of the tumor to healthy tissues, these threshold doses cannot always be respected. This results in deterministic damage: the side effect increases as the absorbed dose increases. These threshold doses are designed to balance the beneficial effects and the side effects of radiotherapy treatment, often associated with chemotherapy. Dose-volume constraints are routinely used in radiotherapy treatment planning through the introduction of normal tissue complication probability (NTCP) models [13]. The development of this field of research in radiotherapy should make it clear how significant it is to establish well-defined dose constraints and to balance the effects that radiotherapy treatments can cause. The concomitant use of systemic therapy with radiotherapy increases the risk of side effects [14]. There are several risk factors that may increase GI toxicity. Some factors are intrinsic to the patient: advanced age, obesity, cigarette smoking, previous surgery and chronic inflammatory bowel disease. Other factors are related to the integrated treatment (for instance, the association of radiation with chemotherapy or other systemic therapies). Radiation-related factors may be due to the type of treatment delivered: the cumulative radiation dose received, the type of fractionation and the type of technology used. Today, linear accelerators allow the delivery of highly conformal radiation treatments that spare healthy organs with high precision, but the proximity of the tumor to the OARs prevents the irradiation of healthy tissue from being totally avoided.

### 4.1. Clinical Presentation of Mucositis of upper Gastrointestinal Tract: Oral Cavity, Esophagus, Stomach

In the oral cavity, mucositis is a side effect that can also be very serious and lead to the interruption of cancer treatments in patients with tumors affecting the head and neck. Several groups have produced classifications to define the grades of oral mucositis. The most widely used is the RTOG classification. Grade I includes mucositis that causes mild pain or congestion and does not require analgesics. Grade II includes the development of patchy mucositis, or those requiring analgesics and those producing serous discharge. Grade III includes the development of confluent mucositis or severe pain requiring narcotic analgesics, while grade IV involves the development of ulceration, necrosis or bleeding from the area. Among other toxicity scales that classify side effects, a widely used one in daily clinical practice is the CTCAE (Common Terminology Criteria for Adverse Events). In this classification, the first sign of mucositis is the erythema, which generally begins toward the end of the second week of chemoradiation. This grade is generally asymptomatic; sometimes the patient may develop initial inappetence or discomfort with eating hot, spicy foods. The progression into grade II involves the development of areas of local desquamation that produce burning symptoms and result in initial dysphagia and dysgeusia. This leads to reduced food intake and causes patient wasting. Grade III involves confluent mucositis with swallowing and speech abnormalities, possible sialorrhea or initial xerostomia. Bacterial and fungal superinfections may occur, which exacerbate symptoms. The development of severe mucositis leads to weight loss and morbidity. This causes fatigue, anemia, anorexia and cachexia in the patients. The development of grade IV mucositis leads to mucosal ulceration, necrosis, and bleeding [15].

Regarding esophageal toxicity, in the acute onset (within 90 days of the end of radiotherapy treatment), typical side effects are dysphagia and odynophagia. In severe cases, bleeding or even esophageal perforation may occur. The effects of delayed toxicity (more than 90 days after the end of radiotherapy treatment) include chronic dysphagia due to ulcers and stenosis. A study conducted on patients with lung cancer using esophago-gastro-duodenoscopy (EGD) showed esophageal toxicity in 42% of treated patients [16]. At the gastric level, if mucositis develops, some effects, such as anorexia, nausea, vomiting, abdominal pain, dyspepsia or bleeding, may occur. Rarely, fibrosis of the wall may lead to obstruction.

### 4.2. Clinical Presentation of Mucositis of Lower Gastrointestinal Tract: Small Bowel, Rectum

The lower gastrointestinal tract may be involved in many radiotherapy treatments of abdominal and pelvic tumors. The small bowel is very sensitive to radiation. Radiation enteritis is a process that is due to inflammation of the mucosa of the small intestine. Mucosal damage can cause epithelial atrophy and reduced blood supply throughout the mucosa. This leads to progressive ischemia, fibrosis and loss of stem cells to regenerate the lost tissue [17]. Regarding the late damage, this is related to the injury and fibrosis of the vessels that supply the mucosa. This causes obliterative endarteritis secondary to radiation, which results in chronic ischemia and fibrosis of the intestinal tract affected by radiotherapy treatment [18]. Acute symptoms typically arise from the second week of radiotherapy treatment and peak in the fourth-fifth week. The most typical manifestations are nausea, abdominal pain, vomiting, bloating, loss of appetite or diarrhea. In severe cases, sepsis secondary to bacterial translocation from the damaged intestinal mucosa can occur [19]. The symptomatology of chronic enteritis is more heterogeneous. It can occur from three months up to years after the end of radiotherapy treatment. Symptoms include chronic abdominal pain, malabsorption (resulting in weight loss), diarrhea, bleeding and complications secondary to stenosis and fibrosis [20]. Proctitis, on the other hand, is the result of damage to the rectal mucosa. Endoscopically, the mucosa may appear erythematous and dropsical, and ulcers may be found [21]. Up to one in five patients receiving radiotherapy treatments in the pelvis develop acute symptoms, such as cramping, nausea, tenesmus, diarrhea, bleeding, mucorrhea and urgency. Late-onset symptoms are similar to those involving the small intestine: they include complications due to fibrotic changes in the tissue, such as fistulas, perforations, obstruction and bleeding [22]. The finding of acute proctitis does not imply the occurrence of chronic proctitis [23]. 

## 5. Preventive Measures and Treatment Modalities

Although oral and intestinal mucositis represent well-known adverse events of antineoplastic treatments, the integration of supportive therapies remains a challenge. The treatment of radiation-induced GI effects depends on the symptoms, the severity of the lesions and localization. Treatment options vary from supportive and dietary measures to specific medical interventions. The prevention and treatment of oral and intestinal mucositis influence patient quality of life and clinical decisions relevant to integrated therapies. The Mucositis Study Group of the Multinational Association of Supportive Care in Cancer/International Society of Oral Oncology (MASCC/ISOO) published clinical practice guidelines on basic care strategies for oral mucositis. Although no guidelines were delineated for patient education and professional oral care due to limited and conflicting evidence, positive results suggest further investigations into the potential benefit of these interventions for oral mucositis management [24]. The recommendation that patient education is an integral part of patient care is supported by two new studies evaluating patient education and quality of life in cancer patients with oral mucositis. Although these studies were not designed to obtain the findings of patient education on oral mucositis prevention, both demonstrated a trend toward a significantly higher quality of life in patients in the education group compared to those in the control group [25,26]. Maintenance of oral hygiene is a relevant issue in the prevention of oral mucositis. Avoiding hot and spicy food, alcohol and smoking is crucial in reducing the incidence and severity of oral mucositis. Mouthwashes with normal saline, soda bicarbonate and benzylamine (non-steroidal analgesic and anti-inflammatory) are effective in the prevention and treatment of radiation-induced oral mucositis. Treatment of coexisting infection is also important and both topical and systemic antifungals should be used. The controls of oral candidiasis may help in improving the nutrition of these patients. As the mucositis progress, opioids may be required starting from a week up to strong opioids (morphine or fentanyl) [27]. For symptomatic relief of pain, non-steroidal anti-inflammatory drugs (NSAID) and proton-pump inhibitors (PPI, in case of distal esophagitis) can be used [28]. Emerging evidence suggests that the gut microbiome may play a role in radiation-induced GI toxicity and that probiotics may have a radioprotective effect on the GI tract [29]. In these last years, many studies investigated if probiotics could prevent chemoradiation-induced diarrhea. In fact, Afifi et al., in a systematic review and meta-analysis, found that probiotics were generally beneficial in treatment-induced diarrhea, especially for grades two and three [21]. However, due to differences in patient characteristics, treatment protocols, dosages and duration of delivery, administration routes, as well as species and strains of probiotics used, comparing the results of the various studies is challenging [30]. Seven out of nine studies showed that probiotics prevent chemotherapy and radiotherapy-induced grade 2 diarrhea and that the co-administration of probiotics with radiotherapy could prevent intestinal toxicity induced by radiotherapy compared with chemotherapy [31]. However, based on current evidence, no precise guidance on how to use probiotics for radiation-induced diarrhea prevention can be given [32]. Regarding the rectum, existing therapeutic strategies for radiation proctitis are mostly extracted from clinical trials and systematized experiences. Medication therapy management for acute symptoms includes oral drugs, clyster and topical medication. Many therapeutic agents are in use, but often with little evidence base. The use of medical therapy (sucralfate, amifostine, sulphasalazine, 5-aminosalicylic acid) to prevent the development of chronic radiation proctitis has only a minimal effect and is not widely used [32,33,34,35].

## 6. The Emerging Protective Role of Natural Compounds on Radiation Injuries due to Antioxidant Activity

Plants have their own radiation protection mechanism that permits them to grow even in the presence of radiation emitted by intense sunlight [4]. The innate radioprotective ability of natural products obtained from plants is in part due to the numerous antioxidants possessed as a part of their normal secondary metabolic processes [4]. To analyze the radioprotective effects of natural compounds on radiotherapy-induced injuries, compounds from five different plant-derived matrices (seeds from grapes (Vitis vinifera L. cv. Italia), leaves of rosemary (*Rosmarinus officinalis* L.), mesocarp of pomegranate fruits (*Punica granatum* L. cv. Wonderful) and OLIVE water obtained from filtered vegetation water from olive milling and Vineatrol^®^30A) were analyzed by using a well-standardized in vitro chemical method (Trolox equivalent antioxidant capacity—TEAC) and an ex vivo biological assay composed by a cellular membrane system obtained from the erythrocytes of healthy volunteers. This assay (ex vivo biological assay) is based on the capability of phytochemical treatment to prevent membrane lipid peroxidation under oxidative stress by UV-B radiation [35]. Pasqualetti et al. demonstrated a high correlation between the phenolic contents of plant-derived compounds and their ability to prevent oxidative injuries in a biological system [35]. In another study, it has been observed that lower doses of herbal preparations show significant advantages in radio-protective properties over toxic doses of synthetic formulations [36]. Moreover, the natural products, when administered before irradiation, have been shown to reduce the deleterious effects of infrared radiation in cells [37]. Many researchers have focused on these natural products as promising radio-protective agents against ionizing radiations [38]. The data presented by Brown et al. (2010) shows that a diet enriched with antioxidants started 24 h after a lethal radiation exposure effectively mitigated death mediated by a sparing of bone marrow cells, perhaps due to a reduction in reactive oxygen species (ROS) in murine models [39]. Moreover, in the same study, a diet containing more antioxidants was shown to be effective at mitigating radiation lethality when it is started 24 h after the radiation exposure, and it is more effective than if given soon after the exposure because natural products act as scavengers against ROS produced after radiation too [39]. In order to prevent RIOM, patients often turn to alternative therapies (i.e., antioxidants). The most known natural product with radio-protective effects is Propolis, also known as “bee glue”, a multi-component hard resin found in beehives. Propolis has been demonstrated to have a positive effect in the alleviation of radiotherapy-induced mucositis, probably due to its antioxidant properties [40], even if the mechanism of mucositis is complex and not completely understood [41,42]. Although the basic principle of the effectiveness of radiotherapy is the oxygen effect, the antioxidant action which we are looking for in the prevention of mucositis does not impact the effectiveness of the treatment. In recent years, however, several studies have shown the emerging role of polyphenols and vitamins as natural substances able to protect against radiation-induced mucositis [43]; in fact, polyphenols’ bioactive compounds can upregulate the mRNAs of many antioxidant enzymes like glutathione peroxidase, catalase and superoxide dismutase, therefore showing a response towards oxidative stress triggered by ionizing radiations [44]. The antioxidant capacity of bioactive molecules and their possible effect, due to dietary intake, on the reduction of negative effects of infrared radiations are described in the following section and resumed in Table 1.

### 6.1. Carotenoids

#### 6.1.1. Beta-Carotene

Beta-carotene, obtained from vegetables and fruits colored orange, yellow and red, is the precursor of vitamin A (retinol). It shows antioxidant activity inhibiting free radical damage to DNA. The beneficial effects of β-carotene are limited, as it is difficult to disperse (being a highly hydrophobic compound) and is highly reactive and unstable to oxygen, which often results in the appearance of degradation products that exhibit pro-oxidant properties [45]. Mills and their colleagues monitored the effect of a diet enriched with beta-carotene in the worsening of oral mucosal damage during radiation and chemotherapy synchronically. Beta-carotene has a protective role on the mucosal membrane within the radiation fields used, although the number of patients studied was small [46]. This effect may be explained by the antioxidants’ ability to scavenge free radicals that are created by the interaction between irradiation and water molecules [47].

#### 6.1.2. Lutein

Lutein is a xanthophyll carotenoid and is one of the carotenoids most present in human serum [48]. It is the di-hydroxyl form of α-carotene [49], and in plants, such as kale and spinach, it is found in dark green leaves [50]. It is especially known to be beneficial for eye health, inhibiting lipid peroxidation with greater effectiveness than β-carotene [51]. Yong et al. examined if the translocation frequency (a biomarker of cumulative DNA damage) of pilots—which are exposed to elevated levels of cosmic radiation—could be associated with intakes of specific carotenoids [52]. They concluded that a high combined intake of vitamins and lutein–zeaxanthin from food, or a diet high in their food sources, could be effective against DNA damage under radiation [52].

#### 6.1.3. Lycopene

In 1876, Millardet first identified lycopene in tomatoes. Lycopene is a linear, unsaturated hydrocarbon carotenoid [53]; it represents the main red pigment of fruits such as tomatoes, pink grapefruit, apricots, blood oranges and watermelon. As a member of the carotenoids, lycopene possesses 11 conjugated double bonds. Its structure, however, is thermolabile and sensitive to oxidative processes [54]. Due to the presence of long chromophorein, the polyene chain yields many red foods [55]. In the presence of other tomato plant constituents, particularly phytofluene, β-carotene and phytoene, its antioxidant activity is synergistically increased [56]. Lycopene has antioxidants and is effective against free radicals on skin cells [57]. Moreover, it is an agent with antioxidant properties often used for radiation-related disorders. Actually, Sadic et al. demonstrated that lycopene exerts radioprotective effects on rat gastrointestinal tract organs, too, especially after a high dose of administration [58]. Motallebnejad et al. investigated if lycopene has an antioxidant effect on the oral mucosa of irradiated rats, and they demonstrated that this natural compound reduced the severity of mucositis and is useful in order to prevent radiotherapy complications (i.e., in head and neck cancers) [59].

### 6.2. Polyphenols

Phenolic compounds have attracted great interest since the 1990s due to growing evidence of their beneficial effect on human health, including anticancer activity [60,61,62]. The phenolic compounds represent a highly heterogeneous group of molecules whose structure is characterized by the presence of a benzene ring bonded to one or more of the hydroxyl groups. They are classified according to their structure into simple, including phenolic acids, such as derivatives of cinnamic acids and benzoic, and complex phenolics, including flavonoids, tannins, and stilbenes. Phenolic compounds represent a highly heterogeneous group of molecules whose structure is characterized by the presence of a benzene ring bonded to one or more of the hydroxyl groups. They are classified according to their structure into simple, including phenolic acids, such as derivatives of cinnamic acids and benzoic, and complex phenolics, including flavonoids, tannins, and stilbenes. The biosynthesis of these compounds occurs through the shikimate pathway and is based on two essential amino acids, phenylalanine and tyrosine, which humans are unable to produce. Therefore, such bioactive molecules must be taken up by human beings through diet [63]. Flavonoids are the most numerous and nutritionally important category of polyphenols and include flavanols, such as catechin and procyanidin; isoflavones, anthocyanins; flavanones and flavones. These compounds generally have two aromatic rings with a phenolic hydroxyl substituent (rings A and B) attached to the carbon moiety [64], containing 15 carbon atoms forming a C6C3C6 structure [65]. These compounds are extensively found in the stem or trunk but also in fruits and, as secondary metabolites, in plants [66,67,68]. Flavonols are found in numerous plants, fruits and vegetables, particularly in red grapes, leeks, tea and apples [69]. In apples, they are predominantly found in the skin [70], while in grapes, they are mostly found in the seeds, both in the monomeric form (+)-catechin, (−)-epicatechin and (−)-epicatechin-3-O-gallate and oligomeric dimeric and polymeric forms [71]. The most common flavonols found in fruit and vegetables are quercetin, kaempferol, fisetin and myricetin [70]. Kaempferol is commonly found in plant-derived foods and in plants used in traditional medicine [72]. It is ingested in glycoside form and absorbed in the small intestine [73]. Quercetin is one of the main flavonoids present in fruits and vegetables such as apples, strawberries [74], onions and broccoli, the latter being considered the best source of bioavailable quercetin [73,74,75]. Stilbenes are taken on in the human diet in low quantities; the resveratrol has been the most representative and extensively studied over the past years. Resveratrol is a polyphenolic flavonoid found in the seeds and skins of grapes, red wine, mulberries, peanuts and rhubarb [69,73,76,77,78]. Resveratrol is biosynthetically correlated to stilbenes, but its biosynthesis is restricted to only a few plant species commonly used for human consumption, among which are pine, peanuts, grapes, bilberry and mulberry [79,80,81]. Different compounds obtained from edible plants, plant by-products and dietary supplements have been compared for their antioxidant properties before and after irradiation of 140 cGy. All the dry compounds, characterized in terms of vitamin C and phenolic content, remained chemically unaltered and maintained their antioxidant capability after irradiation [81]. Polyphenols, epicatechin from green tea, in particular, inhibit apoptosis caused by irradiation, loss of the mitochondrial membrane potential and intracellular ROS-induced injuries in human keratinocytes [82]. In an animal model, after treatment with epicatechin, a significantly decreased amount of radiation-induced apoptotic cells was observed [83]. The antioxidative effects of tea polyphenols were explained by direct radical scavenging, downregulation of radical production, elimination of radical precursors, metal chelation and regeneration of endogenous antioxidants [84,85]. The effect of EGCG on ROS generation could be explained by the protection of mitochondria as a producer and major target of ROS-induced cytotoxicity, and ECG promotes apoptosis and increases the levels of hydrogen peroxide in cancer cells [86,87,88].

#### 6.2.1. Caffeic Acid 

Caffeic acid is a hydroxycinnamic acid derivative and is also known as trans-caffeate or sodium caffeate (CA, (E)-3-(3,4-dihydroxyphenyl) prop-2-enoic acid). Hydroxycinnamic acids, among simple phenolic compounds, are characterized by a cinnamic acid in which the benzene ring is hydroxylated. CA has been isolated from *Ilex paraguariensis*, *Melissa officinalis*, *Baccharis genistelloides* and *Achyrocline satureioides*. It is also present in beverages such as wine, tea, coffee and apple juice [89]. It has numerous pharmacological activities including antioxidant [89,90], antiviral and anti-inflammatory [91]. The antioxidant capacity of CA is due to the delocalization of the split electrons in the conjugated side chain and is enhanced by the formation of additional hydrogen bonds using its ortho-dihydroxyl group after breaking the OH bond. CA also has an antioxidant free radical scavenging activity [92,93,94]. Jin et al. have suggested that pretreatment with caffeic acid could be effective for treating radiation-induced intestinal damage thanks to radioprotective effects on X-ray irradiation-induced gut injury in murine models. After 72 h radiation exposure, a significant reduction in intestinal mucosal apoptosis and oxidative stress was observed in mice pretreated with CA [95].

#### 6.2.2. Gallic Acid

Gallic acid is one of the phenolic acids; it is also called gallate (GA, 3,4,5-trihydroxybenzoic acid). It is found in all living species. Some foods show high concentrations of gallic acid, including mango (*Mangifera indica*) [96], pomegranate (*Punica gran-atum*) [97], cloves (*Syzygium aromaticum*) [98], blueberries and strawberries [99] and grape seeds [100]. The different activities of GA have been attributed to its behavior as both an antioxidant and a pro-oxidant; in fact, GA chelates transition metal ions that cause free radical damage in the human body [101]. Moreover, gallic acid inhibited the peroxidation of membrane lipids leading to lower mortality after γ-irradiation in rats [47].

#### 6.2.3. Ferulic Acid

Ferulic acid (FA, (E)-3-(4-hydroxy-3-methoxyphenyl) prop-2-enoic acid) is a phenolic acid present in fruits and vegetables [102], primarily in leaves and seeds, in free form and covalently bound to lignin and other biopolymers [102]. FA is a strong membrane antioxidant in humans and is noted for its power to be protective against cancer, colds, flu, skin aging and muscle wasting. FA has been demonstrated to be a potent antioxidant that terminates free radical chains [102]. Das et al. has demonstrated, for the first time, the protective action of FA in preventing intestinal damage due to irradiation in animal models [103].

#### 6.2.4. Chlorogenic Acid

Chlorogenic acids (CGAs) are esters of trans-cinnamic acids (e.g., caffeic, ferulic and p-coumaric acids) with (−)-quinic acid. They are classified into different classes according to the nature and number of cinnamic substituents and the position of esterification in the cyclohexanic ring of quinic acid. Caffeoylquinic acid, dicaffeoylquinic acids and feruloylquinic acids are the most abundant CGAs in coffee. These compounds, long known as antioxidants, also slow the release of glucose into the blood stream after a meal. Various scientific evidence shows that CGAs have anti-inflammatory, anti-mutagenic, DNA damage inhibition and antioxidation properties [104,105]. It has been demonstrated that CA is able to reduce mortality in animal models after γ-irradiation exposure [106]. Moreover, CA decreases the DNA damage induced by irradiation in lymphocytes [107].

#### 6.2.5. Cinnamic Acid 

Cinnamic acid (CIA)(E)-3-phenylprop-2-enoic acid), also known as (Z)-cinnamate or 3-phenyl-acrylate, is an organic aromatic compound containing a benzene group and a carboxylic acid forming 3-phenylprop-2-enoic acid. Cinnamic acid is found in several plants, such as *Panax ginseng*, and also in fruits, whole grains, vegetables, honey and cinnamon (*Cinnamomum cassia*) [108]. CIA is a white-colored crystalline organic acid, lightly soluble in water and modestly soluble in many organic solvents. Studies have reported that cinnamic acid shows antioxidant, antimicrobial [109], anticancer [110], neuroprotective, anti-inflammatory and antidiabetic properties [111]. Cinnamic acid terminates radical chain reactions by donating electrons that react with radicals to form stable products [112].

#### 6.2.6. Epigallocatechin Gallate

Epigallocatechin gallate is a flavonoid found in many plants, including *Vitis vinifera*, [(2R,3R)-5,7-dihydroxy-2-(3,4,5-trihydroxyphenyl)-3,4-dihydro-2H-chromen-3-yl] 3,4,5-trihydroxybenzoate). It is able to inhibit cellular oxidation and prevent cell damage due to free radicals. It is under study as a potential cancer chemo-preventive agent [113]. Zhu et al. found that ECGC significantly improved the viability of human skin cells that were irradiated with X-rays and reduced X-ray irradiation-induced apoptosis. Moreover, the same authors have demonstrated that EGCG was safe and able to determine the relief of mucositis symptoms [113]. 

#### 6.2.7. Resveratrol

Resveratrol is a non-flavonoid phenol, a phytoalexin produced naturally by plants in defense against attacks by plant pathogens such as bacteria or fungi (5-[(E)-2-(4-hydroxyphenyl)ethenyl]benzene-1,3-diol). It is present in a few plant species, such as pine, pine peanut, grape, mulberry and blueberry, [81]. This phenolic compound exists in the form of two different geometric isomers, trans- and cis-piceides, but the trans form is the more stable [114]. Resveratrol is a phytoalexin [114], one of the polyphenolic compounds produced by plants in response to environmental stress, such as microbial infections, UV radiation and exposure to ozone [115,116]. Recently, resveratrol has been shown to act as a pleiotropic biological effector, which regulates the multistage carcinogenesis process, and it was found to exhibit multiple bioactivities, including antioxidative, anti-inflammatory, cardiovascular protective and anti-aging properties. Resveratrol was investigated for its effect on growth-inhibitory activity in some human cancer cell lines, and a pro-apoptotic effect was observed in leukemia and mammary and epidermoid cell lines [117]. Firouzi et al. investigated the effect of resveratrol on DNA injuries and colony death on treated glioblastoma cells compared to controls demonstrating a decreased vascular growth in the glioblastoma cells [117]. The hypothesis of this study was that resveratrol leads to the stabilization of p53 by binding to hypoxia inducible factor-1 (HIF1-α) in hypoxic conditions (often found in neoplastic growth), thereby decreasing the function of the vasculogenic cells. They also observed that this natural product detects and kills genetically mutated cells. Moreover, Carsten et al reported that resveratrol decreases the expression of anti-apoptotic proteins like BCL2 and increases the expression of pro-apoptotic proteins like BAX in cancer cells [118]. Thus, it was proved that resveratrol is a promising natural product in treating cell-cycle or ROS-mediated diseases, including radiation-induced cellular damage [117,118]. Resveratrol reduces the impact of radiation on intestinal injury by improving the morphology of the intestine, suppressing crypt cell apoptosis, maintaining regeneration of cells and ameliorating activity and expression of SOD2 as compared to the control. Resveratrol is also able to activate Sirtl and to acetylate p53 expression disrupted via irradiation [44]. One study reported the ineffectiveness of resveratrol and its metabolite piceatannol as chemical radioprotectors as they neither prevented human lung fibroblasts from radiation-induced cell death nor shielded C3H mice against lethal total body IR [44].

#### 6.2.8. Rosmarinic Acid

Rosmarinic acid (RA) (CR, R)-α-[[3-(3,4-dihydroxyphenyl)-1-oxo-2 E-propenyl]oxy]-3,4-dihydroxy-enzenepropanoic acid) is present prevalently in species belonging to the family *Boraginaceae.* RA has been first isolated and identified by two Italian in the plant *Rosmarinus officinalis* from which the name is derived. It is an ester of caffeic acid and 3,4-dihydroxyphenylactic acid and has various astringent, antioxidant, anti-inflammatory, anti-mutagenic, antibacterial and antiviral properties [119]. Fernando et al. have demonstrated, in human keratinocytes, that RA is able to increase the expression and activity of several enzymes usually reduced by UVB radiation—such as superoxide dismutase and catalase. Collectively, these data indicate that RA can provide substantial cytoprotection against the adverse effects of UVB radiation by modulating cellular antioxidant systems, being a potential medical agent for ROS-induced skin diseases [120].

#### 6.2.9. Quercetin

Quercetin (QE, 2-(3,4-dihydroxyphenyl)-3,5,7-trihydroxychromen-4-one) is one of the main flavonoids, possessing five active groups: the dihydroxy group between the A-ring, the o-dihydroxy B-group, the C-ring C2, the C3 double bond and the 4-carbonyl. It is present in fruits and vegetables, including apples, berries, brassica vegetables, tomatoes, capers, grapes, onions, spring onions and tea as well as in many flowers, bark, seeds, nuts and leaves [73,74,75,121,122,123,124,125]. It is a natural polar inhibitor of auxin transport [126]. QE is a potent antioxidant due to the presence of two antioxidant pharmacophores within the molecule that have the optimal configuration for free radical scavenging. The ability to scavenge free radicals makes QE suitable for reducing inflammation [127]. Antioxidants and several naturally occurring plant phenolic glycosides, such as sinapoyl-E-glucoside (sEg), quercetin-3-O-rhamnoside-7-O- glucoside, quercetin-3-O-rhamnoside (q3Or) and luteolin-7-O- (2-apiosyl)-glucoside (l7O2ag), are promising radioprotective agents [128,129]. Arivalagan et al. treated lymphocytes, collected from healthy human donors, with phenolic glycosides before irradiation with X-rays observing that q3Or exhibited a promising radioprotective effect with a 50% reduction in DNA damage compared to controls [128]. Moreover, l7O2ag is reported to possess significant superoxide radical scavenging capabilities, thereby demonstrating its radioprotective potentiality [129]. It has been demonstrated that *Calendula officinalis* extract, rich in antioxidants, especially quercetin, could be effective in decreasing the intensity of radiotherapy-induced oral mucositis (OM). Forty patients with neck and head cancers under radiotherapy or concurrent chemoradiotherapy protocols were randomly assigned to receive either 2% calendula extract mouthwash or placebo; at the end of this study, *Calendula* gel could be effective in decreasing the intensity of radiotherapy-induced OM during the treatment with an antioxidant action partly responsible for the protective effect [130]. To date, current evidence has demonstrated that *Calendula* is effective in decreasing OM intensity but cannot completely prevent its occurrence. Another interesting piece of evidence is the capability of quercetin to significantly reduce lung damage in irradiated mice; following this treatment, an enhanced expression of Ikb-a, an NF-κB inhibitor, was observed with a significantly reduced number of p38, JNK/SAPK and p44/p42 positive inflammatory cells. Based on these results, the radioprotective role of quercetin could be due to the repression of NF-κB and MAPK cascades [44]. Moreover, quercetin decreased skin fibrosis induced by radiation exposure in rats’ irradiated skin as compared to the control group, with a decrease in hind limb contracture, in collagen accumulation, and in the TGF-ß expression [131]. Therefore, it has also been demonstrated, in an animal model of γ-irradiation, that quercetin makes leukocytes more resistant to irradiation by potentiating an adaptive response and significantly prevents DNA damage, specifically when administered before irradiation. In addition, it showed non-toxic effects in non-irradiated mice [44].

#### 6.2.10. Curcumin

Curcumin (1,7-bis(4-hydroxy-3-methoxyphenyl)-1,6-heptadiene-3,5-dione) is a yellow polyphenolic substance also known as diferuloylmethane. Its molecular structure is similar to that of other bioactive non-volatile curcuminoids, such as dimethoxycurcumin and bisdemethoxycurcumin, which differ only in the number of methoxy groups on their aromatic rings. It is extracted from the rhizomes of *Curcuma longa Linn (Zingiberaceae family*) and has long been known as an Indian spice with potent health benefits [132]. This plant has been used since ancient times for both food and pharmaceutical purposes, showing a variety of beneficial effects on the organism reducing glycemia, hyperlipidemia, insulin resistance and steatosis in non-alcoholic fatty liver disease (NAFLD) [131]. More recently, scientific evidence is mounting that curcumin offers innumerable health benefits, all stemming from the fundamental property of decreasing inflammatory mediators [132]. Moreover, curcumin is a completely natural compound with essentially no side effects; tolerance in phase I clinical trials has shown no medically adverse effects for doses up to 8–12 g orally per day [133]. Chikara et al. have shown that curcumin protects normal cells from radiation-induced injuries [134]; curcumin possesses anti-inflammatory effects, probably thanks to the reduction in the production of inflammatory molecules and the increase in the balance between antioxidants and oxidants [133]. Moreover, curcumin has radioprotective properties and is able to sensitize cancer cells to irradiation too [135]. Moreover, Patil et al. have demonstrated the efficacy and safety of a curcumin mouthwash in treating the oral mucositis of patients undergoing radio-chemotherapy compared to chlorhexidine [136]. Moreover, it has been demonstrated that curcumin has a preventative role in the irreversible cerebral damage of 4 Gy carbon ion radiation exposure, decreasing cerebral malondialdehyde (MDA) levels and significantly enhancing the levels of super-oxide dismutase (SOD). The protective impact of curcumin was centered on the initiation of Nrf2-regulated antioxidative cascades. The expression of both Nrf2 and downstream targets, including y-GCS, HO-1 and NQO1, were enhanced in mice pretreated with curcumin [44]. A single oral administration of curcumin (5, 10 and 20 mg/kg b.w.) to mice, either 2 h before or immediately after irradiation, can significantly reduce the production of micronucleated polychromatic erythrocytes (Mn PCEs) induced by whole-body exposure to y-radiation (1.15 Gy; 0.05 Gy/s) [137].

### 6.3. Vitamins

Vitamins are essential components for maintaining a healthy state, so it is important to avoid deficiencies. Vitamins are divided according to their absorption mechanism; the fat-soluble are easily absorbed by fat and the water-soluble are not easily absorbed. The best-known water-soluble vitamins are vitamin C and vitamin B complex, both of which are found in various foods, fruit, vegetables, dairy products, peas, liver, meat, eggs and cereals. In particular, vitamin B complex is important for various processes, such as normal body growth and development, skin health, proper nerve and heart function, and red blood cell formation. Thiamine, riboflavin, nia-cine, pantothenic acid, pyrixodin, biotin, folic acid and cobalamin are all part of the vitamin B complex [138]. Vitamin C (ascorbic acid, ascorbate) is necessary for collagen growth, wound healing, bone formation, enhancing the immune system, absorption of iron and strengthening blood vessels [138,139]. The number of vitamin deficiencies in the head and neck cancer (HNC) population may be numerous. The focus of research conducted by Nejatinamini et al. was vitamins A, D, E, folate and B12. They showed that patients who have diets containing low vitamin content (for example, low plasma levels of 25-OH D and/or all-trans retinol) are more likely to experience mucositis during cancer treatment [140]. Vitamins E, A, C and B have been demonstrated to be promising gastro-intestinal radioprotectors [140]; in fact, gamma-Tocotrienol is a vitamin E that has been demonstrated to confer protection to radiation-induced intestinal damage in vitro [141]. Sayed et al. assessed the impact of pentoxifylline/vitamin E on the incidence and severity of RIOM in head and neck cancer patients, and at the end of the study, pentoxifylline/vitamin E combination reduced the severity and duration of acute radiotherapy-induced oral mucositis [142].

### 6.4. Compounds of a Terpenic Nature

Compounds of a terpene nature are characterized by at least one isoprene unit. They are differentiated by molecular weight and by multiples of their constituent isoprene units: monoterpenes (C10), sesquiterpenes (C15), diterpenes (C20), triterpenes (C30) and tetraterpenes (C40). They are responsible for the fragrance, taste and pigment of plants [143]. Monoterpenes are the most fragrant of the whole terpene family. They come from different flowers, roots (i.e., licorice), fruits and leaves; an example is limonene, which is a monoterpene found in citrus fruits that gives the characteristic aroma or α-pinene which produces the scent of pine. Sesquiterpenes, on the other hand, are important because they are responsible both for the bitter taste, which is nothing more than a defense mechanism against herbivores from feeding on them, and also for the sweet taste or flavors that are pleasing to certain organisms in order to spread their seeds and be fertilized in different areas [144]. Triterpenes are produced by animals, plants and fungi and are known as precursors of steroids in animal and vegetable organisms and derive from mevalonic acid; although they are less known than other terpenic compounds, studies show beneficial effects on human health [145]. Among these compounds, Rocha Caldas et al. demonstrated in vivo that 1.8-cineole (a monoterpene) has antioxidant and gastroprotective activities because it significantly reduces injuries to the mucosal in indomethacin-induced gastric ulcers [146]. A randomized controlled clinical trial assessed the role of licorice mucoadhesive film on head and neck mucositis induced by radiotherapy. They show that licorice mucoadhesive film decreases pain and the level of radiation injuries and could be administered in OM management [147].

**Table 1 nutrients-14-05374-t001:** Nutraceutical properties of natural compounds and their protective effects under irradiation.

	Sources	Properties	Direct and Indirect Effects on Irradiation
**Beta-carotene**	Red, orange and yellow fruits and vegetables [45].	Beta-carotene is a natural precursor of retinol (vitamin A) [45].	It scavenges free radicals, which belong to the interaction between irradiation, and it significantly reduces the number of double-strand breaks in peripheral blood mononuclear cells of patients undergoing 99mTc methylene diphosphonate bone scans compared to the control group [47].
**Lutein**	Dark green leafy vegetables such as spinach and kale [50].	Lutein is the second most prevalent carotenoid in human serum [49], has two hydroxyl groups, one on each side of the molecule, and is the dihydroxy form of α-carotene [49].	It protects against cumulative DNA damage in IR-exposed persons [52].
**Lycopene**	The main red pigment of fruits, such as tomatoes, pink grapefruit, apricots, blood oranges and watermelon [48].	Lycopene is a linear, unsaturated hydrocarbon carotenoid first discovered in tomatoes by Millardet in 1876. It is an acyclic carotene with 11 conjugated double bonds [48].	Reduces the severity of mucositis and can be used to prevent RIOM, especially in the treatment of head and neck cancers [59].
**Caffeic Acid**	*Ilex paraguariensis*, *Melissa officinalis*, *Baccharis genistelloides* and *Achyrocline satureioides*. It is present in wine, tea, coffee and apple juice [89].	It belongs to the class of organic compounds known as hydroxycinnamic acids [89].	Radioprotective effects against X-ray irradiation-induced intestinal damage in rats [94]. It also reversed the activation of p38MAPK and the increased expression of inter-cellular cell adhesion molecule-1 induced by radiation in the intestinal mucosa [95].
**Gallic Acid**	Mangos (*Mangifera indica*), pomegranates (*Punica granatum*), blueberries and strawberries, grape seeds and cloves [96,98,99].	It belongs to the class of organic compounds known as gallic acids [98].	GA has an antioxidant and pro-oxidant activity. GA is able to chelate transition metal ions that are promoters of free radical damage in the human body [101].
**Ferulic Acid**	Fruits and vegetables [102].	It belongs to the phenolic acid group commonly found in plant tissue [139] that arises from the metabolism of phenylalanine and tyrosine [102].	Is a potent antioxidant and terminates free-radical chains [102]. It has a protective action in preventing intestinal injury due to γ-radiation [103].
**Chlorogenic Acid**	Coffee, leaves and fruits of dicotyledonous plants [104].	It is a cinnamate ester obtained by formal condensation of the carboxy group of trans-caffeic acid with the 3-hydroxy group of quinic acid [104].	Anti-inflammatory and antioxidant [103], anti-mutagenic, DNA damage inhibition and antioxidation properties [105].
**Cinnamic Acid**	It is present in fruits, whole grains, vegetables and honey [108].	It belongs to the class of organic compounds known as cinnamic acids. These are organic aromatic compounds containing a benzene group and a carboxylic acid forming 3-phenylprop-2-enoic acid [108].	Antioxidant, antimicrobial [109], anticancer [110] and anti-inflammatory [111]. Cinnamic acid terminates radical chain reactions by donating electrons that react with radicals to form stable products [112].
**Epigallocatechin gallate**	Several plants such as green tea and in *Vitis vinifera* grape seeds [113].	EGCG, with 8 hydroxyl groups, notably in 3′, 4′ and 5′ positions and with a gallate moiety in C-3, is a better electron donor than the other catechins and thus the best scavenger of free radicals species [148,149].	Relief of the mucositis symptoms [113].
**Resveratrol**	Mulberries, peanuts and grapes [114].	It is a phytoalexin, a polyphenolic compound produced by plants in response to environmental stress, UV radiation, etc. [115]. Resveratrol exists as two geometric isomers, trans- and cis-, and their glucosides, trans- and cis-piceids [115].	It is a promising natural product in treating cell cycle or ROS-mediated diseases, including radiation-induced cellular damage [115,116,117,118].
**Rosmarinic Acid**	*Rosmarinus officinalis,* Boraginaceae family, subfamily Nepetoideae [119].	It is an ester of caffeic acid and 3,4-dihydroxyphenylactic acid [119].	It increases the expression and activity of antioxidant enzymes in human keratinocytes [120]. It is a cyto-protector against the adverse effects of UVB radiation by modulating cellular antioxidant systems [120].
**Quercetin**	Apples, berries, brassica vegetables, capers, grapes, onions, spring onions, tea, tomatoes, seeds, nuts, bark and leaves [121].	Its molecular formula is C_15_H_10_O_7_. It’s a naturally occurring polar auxin transport inhibitor. It has a ketocarbonyl group in its molecules, and the oxygen atom on the first carbon is basic and can generate salts with strong acids. Its molecular structure contains five active groups, namely a dihydroxy group between the A ring, o-dihydroxy group B, C ring C2, C3 double bond and 4-carbonyl [126].	Significant superoxide radical scavenging capabilities [128,129]. It could be effective on the decreasing the intensity of RIOM [130].
**Curcumin**	Rhizomes of plants of Curcuma longa Linn (Zingiberaceae family) [131].	It’s also known as diferuloylmethane. It’s a polyphenolic yellow substance. It has a structure similar to other bioactive non-volatile curcuminoids, such as dimethoxy-curcumin and bisdemethoxy-curcumin, differing only for the number of methoxy groups on their aromatic rings [132].	It protects normal cells from radiation-induced damage [133]. It has anti-inflammatory effects [132], useful in the treatment of oral mucosal ulceration [135].
**Vitamins**	Vegetables and fruits, as well as dairy, meat, legumes, peas, liver, eggs and fortified grains and cereals [138].	There are two main groups of vitamins—fat-soluble (easily stored in fat upon absorption) and water-soluble (washed out and not easily stored). Although adequate intake of all vitamins is important, regular intake is required to avoid deficiency due to the transient nature of water-soluble vitamins [138].	Vitamins E, A, C and B are gastro-intestinal radioprotectors [138]. They confer protection against radiation-induced intestinal damage in vitro [138]. Pentoxifylline/vitamin E combination reduced the severity and duration of acute radiotherapy-induced oral mucositis [142].
**Terpenes**	Flowers, roots (i.e., licorice), fruits and leaves [144].	Are characterized by at least one isoprene unit. They are differentiated by molecular weight and by multiples of their constituent isoprene units: monoterpenes (C10), sesquiterpenes (C15), diterpenes (C20), triterpenes (C30) and tetraterpenes (C40) [143].	Antioxidant and gastroprotective activities [146]. Their application decreases pain and the level of radiation injuries [147].

## 7. Conclusions

In this review, an interesting synthesis of several natural products with protective properties possibly effective in the prevention of RIOM is proposed. As demonstrated, these substances are natural antioxidant compounds that, when administrated before irradiation, have been shown to reduce deleterious effects of infrared radiations in cells and cellular injury mediated by inducing the expression of genes that encode detoxifying enzymes.

Numerous in vitro studies and animal model investigations have shown that several natural products may protect against cumulative DNA damage and intestinal injuries after IR exposition, prevent radiotherapy complications (especially in the treatment of head and neck cancers) and that these substances are able to determine the relief of the mucositis symptoms.

Unfortunately, it is still not possible to determine and evaluate what would be the recommended doses (e.g., in mg/day) of natural compounds able to control and reduce mucositis symptoms due to the lack of enough evidence; however, it would be interesting to determine an evaluation of recommended doses to create nutraceutical supplements in order to alleviate radiotherapy injuries, especially mucositis.

A limitation of the current literature regarding this topic is also due to the reduced number of preclinical models of radiotherapy mucositis, which does not permit to confirm the preventive effect of natural products in this pathological condition. Further studies are needed to better understand the mechanism underlying radiation-induced mucositis and the potential benefit of natural antioxidant substances for oral mucositis prevention and management.

## Data Availability

Not applicable.

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
