# Peer review of "Protective Role of Natural Compounds under Radiation-Induced Injury"

_nutrients, 2022, doi:10.3390/nu14245374_

Round 1

Reviewer 1 Report

1. A review article: "Radiation-induced mucositis and the emerging protective role of natural extracts: up to date" contains substantial material (chapters 2,3,4,5) that is indirectly relevant to the journal's aims and scope. It would be more appropriate to include the contents of these chapters in an article for publish in journal Cancer, for example.

2. Given the profile of the journal, one can recommend that the content of these chapters be shortened, adapted for a wide range of readers, and the reduced material included in a single chapter.

3. The title of the article should also be changed. For example: "Protective role of natural compounds under radiation-induced injury".

4. The name natural extracts should be more correctly replaced with compounds, since the review refers not only to extracts, but also to other compounds, such as propolis.

5. Since a review article is presented, the methodology of searching of information sources for the review should be mentioned.

6. Figures 1 and 2 should better be excluded, because they are of interest to specialists, but have minor relation to the focus of the journal and the radioprotective properties of natural compounds.

7. The title of Table 1 would be clearer if it were replaced with "Nutraceutical properties of natural compounds and thier protective effects under irradiation"

8. The repetition of "red wine" in line 369 need remove.

9. The conclusion is devoted to a discussion of problems in patients with mucositis, and not to the protective properties of natural compounds. This contradicts the main aim and scope of the journal. The conclusion should discuss the prospects of finding new and using already known natural compounds with radioprotective properties to reduce the adverse effects of radiotherapy treatment.

Author Response

Dear Reviewer 1, 

Thank you for your suggestions. Please find attached the answers to your comments. 

Point 1: A review article: "Radiation-induced mucositis and the emerging protective role of natural
extracts: up to date" contains substantial material (chapters 2,3,4,5) that is indirectly relevant to the
journal's aims and scope. It would be more appropriate to include the contents of these chapters in
an article for publish in journal Cancer, for example.
Response 1: We thank the reviewer for this comment.We have shortened the chapters indicated. We
will certainly consider his/her suggestion for a future work in another journal.
Point 2: Given the profile of the journal, one can recommend that the content of these chapters be
shortened, adapted for a wide range of readers, and the reduced material included in a single chapter.
Response 2: We thank the reviewer for this comment. We thank the reviewer for his/her attention to
the text and content. We have shortened the text as suggested, removing parts that might be too
detailed. We have preferred to keep the division into paragraphs to make clearer to a wide range of
readers the background and the clinical need in which the topic of the paper fits.
Point 3: The title of the article should also be changed. For example: "Protective role of natural
compounds under radiation-induced injury".
Response 3: We thank the reviewer for this comment. We have changed the title, as suggested.
Point 4: The name natural extracts should be more correctly replaced with compounds, since the
review refers not only to extracts, but also to other compounds, such as propolis.
Response 4: We thank the reviewer for this comment. We have replaced extracts with compounds
(lines 24-26-43-310-360-370-649), as suggested.
Point 5: Since a review article is presented, the methodology of searching of information sources for
the review should be mentioned.
Response 5: We thank the reviewer for this comment. We have mentioned methology of searching
from line 44 to 54.
Point 6: Figures 1 and 2 should better be excluded, because they are of interest to specialists, but have
minor relation to the focus of the journal and the radioprotective properties of natural compounds.
Response 6: We thank the reviewer for this comment. We have excluded figures 1 and 2 and we also
cancelled their mention in the text (lines 147 and 210).

Point 7: The title of Table 1 would be clearer if it were replaced with "Nutraceutical properties of
natural compounds and their protective effects under irradiation".
Response 7: We thank the reviewer for this comment. We have modified the title of Table 1 as
suggested at line 1282.
Point 8: The repetition of "red wine" in line 369 need remove.
Response 8: We thank the reviewer for this comment. We have removed “red wine” repetition at line
643.
Point 9: The conclusion is devoted to a discussion of problems in patients with mucositis, and not to
the protective properties of natural compounds. This contradicts the main aim and scope of the
journal. The conclusion should discuss the prospects of finding new and using already known natural
compounds with radioprotective properties to reduce the adverse effects of radiotherapy treatment.
Response 9: We thank the reviewer for this comment. We have modidy the conclusion to not
contradicts the main aim and scope of the review (from line 1311 to 1332) as suggested.

Reviewer 2 Report

Dear Authors,

I have found this paper interesting and well written.

The content is a good review of the major natural compounds useful to counteract the radiotherapy-induced mucositis which is a serious side effects for patients.

The literature is well described as well as the relevant biochemical pathways for the action of these compounds.

Meanwhile I have 2 requests:

1/ It seems to me that 2 other families of compounds which have not been addressed could have interesting effects in the field: terpenes/essential oils and triterpenes and which also may be promising for the future.

A few examples of papers on radiotherapy/chemotherapy-induced mucositis:

Terpenes and EOs

Hou Q, Liu L, Dong Y, Wu J, Du L, Dong H, Li D. Effects of Thymoquinone on radiation enteritis in mice. Sci Rep. 2018 Oct 11;8(1):15122. doi: 10.1038/s41598-018-33214-3. Erratum in: Sci Rep. 2021 Mar 24;11(1):7151

https://ajps.uomustansiriyah.edu.iq/index.php/AJPS/article/view/625

Nigella Sativa Oil for Oral Mucositis

Rocha Caldas GF, Oliveira AR, Araújo AV, Lafayette SS, Albuquerque GS, Silva-Neto Jda C, Costa-Silva JH, Ferreira F, Costa JG, Wanderley AG. Gastroprotective Mechanisms of the Monoterpene 1,8-Cineole (Eucalyptol). PLoS One. 2015 Aug 5;10(8):e0134558. doi: 10.1371/journal.pone.0134558

Wu J, Gan Y, Li M, Chen L, Liang J, Zhuo J, Luo H, Xu N, Wu X, Wu Q, Lin Z, Su Z, Liu Y. Patchouli alcohol attenuates 5-fluorouracil-induced intestinal mucositis via TLR2/MyD88/NF-kB pathway and regulation of microbiota. Biomed Pharmacother. 2020 Apr;124:109883. doi: 10.1016/j.biopha.2020.109883. Epub 2020 Jan 28

Araruna ME, Serafim C, Alves Júnior E, Hiruma-Lima C, Diniz M, Batista L. Intestinal Anti-Inflammatory Activity of Terpenes in Experimental Models (2010-2020): A Review. Molecules. 2020 Nov 20;25(22):5430. doi: 10.3390/molecules25225430.

EÄŸilmez OK, Kökten N, KalcıoÄŸlu MT, Ekici AID, Åžerifler S, YeÅŸilada E. Investigation of the Protective Effect of Nigella Sativa Oil in Cisplatin Induced Oral Mucositis: An Experimental Study. Turk Arch Otorhinolaryngol. 2020 Mar;58(1):10-15. doi: 10.5152/tao.2020.4733. Epub 2019 Sep 2

Hussain SA, Mohammed Ameen HA, Mohammed MO, Ahmed KM, Hama-Gareb Ali R, Safar BM, Saeed KA. Nigella sativa Oil Mouth Rinse Improves Chemotherapy-Induced Oral Mucositis in Patients with Acute Myeloid Leukemia. Biomed Res Int. 2019 Oct 30;2019:3619357. doi: 10.1155/2019/3619357.

Triterpenes

Pakravan F, Salehabad NH, Karimi F, Isfahani MN. Comparative Study of the Effect of Licorice Muco-adhesive Film on Radiotherapy Induced Oral Mucositis, A Randomized Controlled Clinical Trial. Gulf J Oncolog. 2021 Sep;1(37):42-47

Somensi LB, Costa P, Boeing T, Mariano LNB, Longo B, Magalhães CG, Duarte LP, Maciel E Silva AT, de Souza P, de Andrade SF, da Silva LM. Gastroprotective properties of Lupeol-derived ester: Pre-clinical evidences of Lupeol-stearate as a potent antiulcer agent. Chem Biol Interact. 2020 Apr 25;321:108964. doi: 10.1016/j.cbi.2020.108964. Epub 2020 Jan 29

Please add a paragraph or discuss your opinion about that.

2/ Is it possible to determine an evaluation of what would be the recommended doses (in mg/day for ex) of the compounds cited in the study to have possible significant effects on mucositis ?

Best regards.

Author Response

Dear Reviewer 2, 

Thank you for your suggestions. Please find attached the answers to your comments. 

Response to Reviewer 2 Comments
Point 1: It seems to me that 2 other families of compounds which have not been addressed could have
interesting effects in the field: terpenes/essential oils and triterpenes and which also may be
promising for the future.
Please add a paragraph or discuss your opinion about that.
Response 1: We thank the reviewer for this comment. We have added a paragraph to describe these
compounds and their activity in this field (from line 1234 to 1279) and we have inserted them into
Table 1.
Point 2: Is it possible to determine an evaluation of what would be the recommended doses (in
mg/day for ex) of the compounds cited in the study to have possible significant effects on mucositis ?
Response 2: We thank the reviewer for this comment. Unfortunately is not possible to determine the
recommended doses of the compounds cited on mucositis because still not exist general guidelines
and enough studies. We have added this explanation in Conclusions, from line 1322 to 1326.
